# Milking Reactivity in Primiparous Saanen Goats During Early Lactation: Effects on Milk Yield, Milk Quality and Plasma Cortisol Concentration

**DOI:** 10.3390/ani14233365

**Published:** 2024-11-22

**Authors:** Mayara Andrioli, Joseph K. Grajales-Cedeño, Aline C. Sant’Anna, João A. Negrão, Mateus J. R. Paranhos da Costa

**Affiliations:** 1Graduate Program in Animal Science and Research Group in Ethology and Animal Ecology, Faculty of Agricultural and Veterinary Sciences, UNESP, Jaboticabal 14884-900, SP, Brazil; may.andrioli6@gmail.com (M.A.); joseph.kaled@unesp.br (J.K.G.-C.); 2Department of Animal Science, Faculty of Agricultural Sciences, University of Panama, Ciudad de Panama 7096, Panama; 3Department of Animal Science and Research Group in Ethology and Animal Ecology, Faculty of Agricultural and Veterinary Sciences, UNESP, Jaboticabal 14884-900, SP, Brazil; aline.santanna@unesp.br; 4CNPq—Conselho Nacional de Desenvolvimento Científico e Tecnológico, Brasilia 71605-001, DF, Brazil; 5Department of Basic Sciences, Faculty of Animal Science and Food Engineering, University of São Paulo, Pirassununga 13635-900, SP, Brazil

**Keywords:** cortisol, milk fat, milk protein, somatic cell count, temperament

## Abstract

At the beginning of the lactation period, primiparous goats are likely to face stressful situations during milking procedures that may negatively affect milk yield and quality. The milking reactivity of 31 primiparous Saanen goats was assessed on days 10, 30, and 60 of the lactation period by scoring step-kick behavior (SK) and head, ear, and tail movements (MOV) during the milking procedure. Milk yield, milk protein and fat percentages, somatic cell count, and plasma cortisol concentrations were measured on day 10 of lactation. Goats’ milking reactivity was classified as low or high, according to the SK and MOV scores measured on day 10 of the lactation period. There was a decrease in the SK and MOV scores over the lactation period. Milking reactivity affected only milk fat percentage, with the milk of goats with low reactivity having more milk fat than those with high reactivity. There was no evidence of stress when goats were subjected to milking procedures early in lactation. Further studies are needed to clarify the relationships of milking reactivity with stress response and their effects on milk yield and quality.

## 1. Introduction

There is a growing interest in animal temperament and how it can affect farm management and daily work routines. Temperament, defined as inter-individual differences in animals’ behavioral responses that are consistent over time and across situations [1], is a complex trait that involves a combination of different dimensions or aspects that affect the expression of various behaviors, making it difficult to measure [2].

Studies of goat temperament are limited [3] and have focused on the dimensions of exploration, sociability, and aggression [4,5]. A study conducted three decades ago showed individual differences in goat temperament when exposed to different human–animal interactions at an early age, and that such differences were later associated with the inhibition of milk ejection [6]. In a more recent study, Sramek et al. [7] compared the milking temperament and udder health of horned and polled Alpine goats and reported that polled goats were calmer and had healthier udders than horned goats.

One of the temperament traits that directly affects handling efficiency is reactivity, defined as a behavioral expression during handling that is generally attributed to fear and is associated with stimuli elicited by human presence [8]. Therefore, one way to assess the temperament of goats is to measure their reactivity during certain types of handling, such as milking, which can be affected by individual differences [9]. In studies with dairy cattle, reactivity has been assessed during milking procedures by assigning visual scores on predefined scales based on animals’ reactions during udder preparation or teat cup attachment [10,11]. This assessment allows us to identify animals with greater behavioral reactivity to handling, which are more susceptible to stress during milking. In this context, the study by Marçal-Pedroza et al. [12] showed that more reactive cows have higher plasma cortisol concentrations. Furthermore, reactivity may be related to milk yield, as reported by Hedlund and Løvlie [13], who observed that nervous cows produce less milk. Milk quality can also be affected, as shown in the study by Marçal-Pedroza et al. [12], which showed that more reactive cows had a higher fat content in their milk and a tendency to lower protein content compared to cows with intermediate reactivity.

Despite the potential risk that high milking reactivity in primiparous goats may lead to stressful situations that harm animal welfare and milking performance, we did not find any studies that examined these issues. These potential effects of high reactivity are particularly relevant at the beginning of lactation when naïve goats are faced with a challenging situation, before adapting to the new environment and handling procedures. Therefore, the aim of this study was to investigate the progression of milking reactivity during lactation and its effects at early lactation on milk yield, milk quality, and plasma cortisol concentration in primiparous Saanen goats. We hypothesized that the more reactive the goats are in the milking parlor, the lower the milk yield and quality, and the higher the plasma cortisol concentration.

## 2. Materials and Methods

This study was approved by the Committee of the Ethical Use of Animals of the Faculty of Agricultural and Veterinary Sciences, São Paulo State University (UNESP), Jaboticabal, SP, Brazil (protocol number 2036/21).

### 2.1. Study Location and Animals

The study was carried out at the Animal Physiology Laboratory of the Faculty of Animal Science and Food Engineering, University of São Paulo, in Pirassununga, SP, Brazil, evaluating thirty-one primiparous Saanen goats at days 10, 30, and 60 of the lactation. Milking took place in a side-by-side milking parlor with a capacity of 12 goats, with 6 of them milked simultaneously with a vacuum level of 48 kPa and a pulsating frequency of 120 cycles per minute. The milking was performed once a day, starting at 7:30 a.m. and ending around 9:30 a.m.

Immediately after milking, we drove the goats to a paddock (Tifton 85, *Cynodon dactylon* (L.) Pers.), with free access to water and mineral supplements. At around 3:00 p.m., we transferred them to a collective pen, where they remained until the following morning, being fed a concentrate (composed of corn grain, soybean meal, minerals, and vitamins) and corn silage, with free access to water and mineral supplements. The diet was formulated according to the NRC recommendations [14], offering an amount of feed to all goats that maintained a surplus of at least 10%, adjusted monthly based on the goat’s weight and lactation stage.

### 2.2. Milking Reactivity

Step-kick behavior (SK) and head, ear, and tail movements (MOV) were assessed by assigning scores, as described in Table 1. Assessments were performed by a previously trained observer. Data collection was carried out three times during pre-dipping and teat cup placement on days 10, 30, and 60 of the lactation period, with measurements recorded three times each day. The daily averages were calculated and used in data analyses.

The goats were classified according to their milking reactivity on the first evaluation (day 10), defining two classes: low reactivity (when SK scored 1, 2, and 3, and MOV scored 1 and 2, *n* = 12) and high reactivity (when SK scored 4, and MOV scored 3, *n* = 19).

### 2.3. Milk Yield and Quality

The milk yield, milk protein and fat percentages, and somatic cell count of each goat were recorded on day 10 of the lactation period. Daily milk yield was measured (kg/day) with the milking equipment (Westfalia, GEA Brasil, Campinas, SP, Brasil). Individual milk samples were collected in 50 mL plastic containers. Milk protein and fat content percentages were measured immediately after milking using ultrasonic equipment (MilkoScope Expert, Razgard, Bulgaria). Milk samples for SCC were frozen and later analyzed by using a microscopic method [16], being transformed into the logarithmic [Log2(SCC × 10 − 5) + 3] somatic cell score (SCS), according to Ali and Shook [17].

### 2.4. Plasma Cortisol Concentration

Blood samples were obtained by jugular venipuncture into 5 mL vacuum tubes containing an anticoagulant (heparin). The samples were centrifuged (15 min, 3250 rpm), and the plasma was stored at −30 °C. Blood collections were carried out on day 10 of the lactation period, after morning milking (~10 a.m.). Plasma cortisol concentration was measured using an enzyme immunoassay kit (Cortisol Test System, Monobind Inc., Lake Forest, CA, USA).

### 2.5. Statistical Analysis

The statistical analyses were performed in R software with the RStudio integrated development environment (R version 4.3.0 (2023-04-21). Firstly, we estimate the Spearman correlation coefficients between SK—pre-dipping and SK—teat cup attachment, and between MOV—pre-dipping and MOV—teat cup attachment. To assess the progression of SK and MOV over the lactation period, we used a linear mixed model using the “glmer” package [18], adjusted for Poisson distribution, considering milking reactivity as the response variable, and lactation (days 10, 30, and 60) as a fixed effect). The animal was included as a random effect. The impact of the explanatory variables was calculated using the “anova” function, running Type III Wald chi-square trials. The best fit of the model was performed with the ‘step-up’ procedure by the Akaike information criterion (AIC) and Bayesian information criterion.

Milk yield and quality (milk fat and protein percentages, and SCS) and plasma cortisol concentration data were analyzed using generalized linear models, considering the classifications of milking reactivity (low vs. high) as a fixed effect and the animal as a random effect. The best fit for all models was performed with the ‘step-up’ procedure using the Akaike information criterion (AIC) and Bayesian information criterion (BIC). Multiple comparisons for the adopted models were performed using the Tukey test to compare the adjusted means.

## 3. Results

Significant positive correlations were found between SK (pre-dipping and teat cup attachment: rs = 0.82; *p* < 0.001) and MOV (pre-dipping and teat cup attachment: rs = 0.92; *p* < 0.001). Based on these results, we considered only SK—pre-dipping and MOV—pre-dipping for the subsequent data analyses.

### 3.1. Progression of Milking Reactivity over the Lactation Period

Figure 1 shows the distribution of the number of animals according to the SK and MOV scores recorded over the lactation period. On day 10 of lactation, a high number of animals scored SK = 4 and MOV = 3, which decreased in subsequent assessments, resulting in a high percentage of animals scoring 1 for both (SK and MOV) on day 60 of the lactation period.

There were significant effects of the lactation period on SK (χ^2^ = 17.11, *p* < 0.001) and MOV (χ^2^ the = 31.63, *p* < 0.001). Figure 2 shows the means of SK and MOV over the lactation period.

### 3.2. Milk Yield and Quality

No effect of milking reactivity was observed on milk yield (F = 0.22, df = 1; *p* = 0.64), milk protein percentage (F = 3.75, df = 1; *p* = 0.06), and somatic cell count (F = 0.05, df = 1; *p* = 0.85), as shown in Figure 3a,c,d, respectively. Milking reactivity had a significant effect on the milk fat (F = 6.51, df = 1; *p* = 0.02), with goats exhibiting low reactivity producing milk with a higher milk fat percentage (Figure 3b).

### 3.3. Plasma Cortisol Concentration

No effect of milking reactivity was observed on plasma cortisol concentration (F = 0.32, df = 1; *p* = 0.57), but an individual variation on this variable was noted, as indicated by the circles in Figure 4.

## 4. Discussion

A significant reduction in reactivity to milking throughout the lactation period confirms that primiparous goats become habituated to the milking process, as previously observed in dairy cows [19]. This habituation is essential to reduce detrimental effects on animal welfare and productivity, as well as to optimize management and ensure handler safety [20,21]. Although the results showed a significant reduction in reactivity throughout the lactation period, individual differences were observed and should be considered during milking management, as described in Holstein × Gir cows [9] and Angus × Hereford cattle [22].

We hypothesized that goats with low reactivity would produce more milk than highly reactive goats, based on studies with dairy cattle showing that more reactive cows produce less milk [23,24,25]. However, in the present study with goats, we did not observe significant differences in milk yield among the different reactivity levels in Saanen goats. This result corroborates those reported by Marçal-Pedroza et al. [12], who also found no variation in milk yield related to the milking reactivity of dairy cows. The authors [12] conducted additional tests to assess cow temperament and reported differences in milk yield associated with the time taken by the animals to enter the corral. Cows with an intermediate temperament produced more milk than those classified as calm or reactive. In the present study, we were limited to temperament assessments in the milking parlor (milking reactivity) and a small sample size. Therefore, further studies involving additional temperament traits and a larger number of animals are needed to clarify the effects of goats’ reactivity on milk yield.

The less reactive goats produced milk with a higher fat content than the highly reactive ones, but no differences were found in the milk protein content. Kruszynski et al. [26] reported similar results, showing that calmer cows produced milk with a higher fat content. However, in contrast to our results, the authors also reported high protein content. In contrast, Marçal-Pedroza et al. [12] showed that calmer cows produced milk with lower fat and higher protein content, while Morales-Piñeyrúa et al. [27] reported lower protein and fat content in calm cows, and Cziszter et al. [28] reported higher fat content in the milk produced by more agitated cows compared to that from cows with intermediate temperament, which, in turn, had lower protein content than milk from calmer and more agitated cows. Finally, Orbán et al. [29] did not identify a significant effect of cows’ reactivity on protein and fat content in the milk.

These results highlight the lack of consensus in the literature regarding the impact of animal reactivity on milk quality. Although our results showed that the less reactive goats produced milk with a higher fat content, they should be interpreted with caution due to the small sample size. Therefore, future studies with a larger number of animals are needed to confirm the differences in milk quality associated with milking reactivity.

Regarding SCS, no differences were observed between the milking reactivity classes. Despite this lack of significance, we hypothesized that the high SCS scores showed by some highly reactive goats probably result from their behavior during milking, as they make the milking process more difficult, potentially spreading more microorganisms through stomping and jumping. Further, studies addressing the effects of milking reactivity on somatic cell count in goat’s milk are needed to confirm this hypothesis.

The classes of milking reactivity also did not affect the plasma cortisol concentration, corroborating the results of Sutherland and Huddart [10], Sutherland et al. [11], and Van Reenen et al. [30], who assessed dairy cows reactivity using reactivity scores similar to ours and did not find any association with plasma cortisol concentration. In contrast, Marçal-Pedroza et al. [12], Wenzel et al. [31], and Gygax et al. [32] reported different results from ours, showing that more reactive cows produced milk with higher cortisol concentrations than the calm ones. Discrepancies in cortisol sampling methods may explain these differences, as assessing cortisol in milk is a non-invasive method that does not cause the additional stress associated with blood collection. However, the plasma cortisol concentration values found in the goats in our study were considered low compared to those in other studies, such as those involving dehorning [33,34], and therefore do not indicate physiological stress in these animals.

Finally, due to the novelty and the limitations of this study, we strongly recommend carrying out further studies aiming to assess the effect of milking reactivity on milk yield and quality, considering a larger number of animals and expanding the range of reactivity traits assessed.

## 5. Conclusions

The goats reduced milking reactivity over the lactation period, indicating habituation to the milking procedures. Regardless of their reactivity, the assessed goats did not show signs of stress when subjected to milking procedures early in lactation. Milk fat percentage was the only milk quality trait affected by milking reactivity. Therefore, we reject the hypothesis that the highly reactive primiparous goats have a higher plasma cortisol concentration and somatic cell count and lower milk yield when subjected to milking procedures.

## Figures and Tables

**Figure 1 animals-14-03365-f001:**
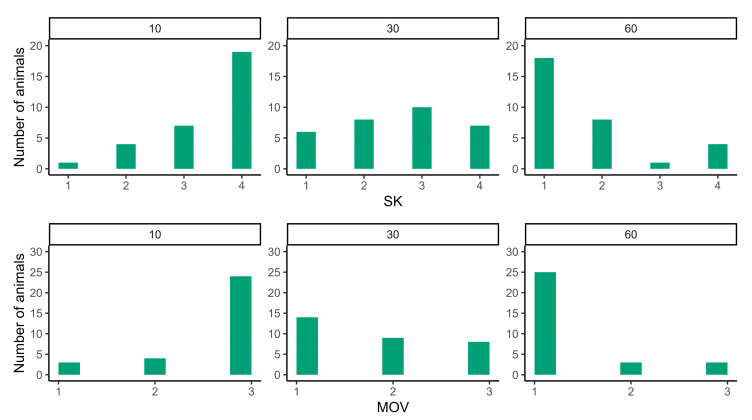
Distribution of the number of animals (n = 31) according to step-kick (SK) and head, ear, and tail movement (MOV) scores during pre-dipping procedures according to the lactation period (days 10, 30, and 60 of the lactation period).

**Figure 2 animals-14-03365-f002:**
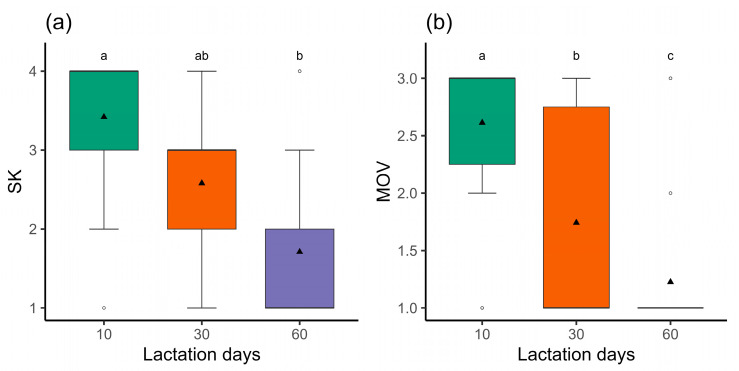
Box plots of (**a**) the step-kick behavior scores (SK) and (**b**) head, ear, and tail movement (MOV) scores in primiparous Saanen goats (*n* = 31) according to the lactation period (days 10, 30, and 60). Different letters indicate significant differences between the assessment days (*p* < 0.05). Bold lines indicate medians, triangles represent means, and circles the outliers.

**Figure 3 animals-14-03365-f003:**
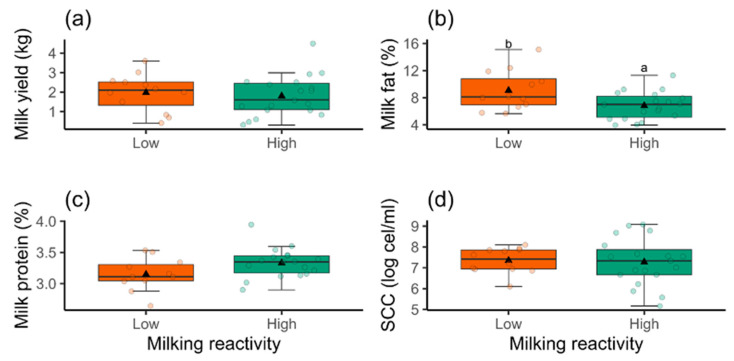
Box plots of milk yield, kg (**a**), milk fat, % (**b**), milk protein, % (**c**), and somatic cell score (log cel/mL) (**d**) in primiparous Saanen goats (*n* = 31) according to the classes of milking reactivity (low and high reactivity). Different letters indicate significant differences between the milking reactivity classes (*p* < 0.05). Bold lines indicate medians, triangles represent means, and circles the individuals.

**Figure 4 animals-14-03365-f004:**
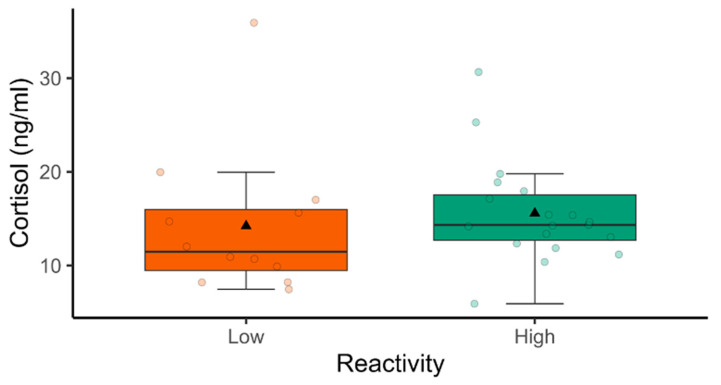
Box plots of plasma cortisol concentration (ng/mL) in primiparous Saanen goats (*n* = 31) according to the classes of milking reactivity (low and high). Bold lines indicate medians, triangles represent means, and circles the individuals.

**Table 1 animals-14-03365-t001:** Descriptions of step-kick behavior (SK) and head, ear, and tail movement (MOV) scores assessed during milking procedures.

SK Scores *	Descriptions
1	The goat is still, does not move any of its legs, shows relaxation in response to milker contact, or remains with its legs stretched. It may also arch its spine or spread its legs when in contact with the milker.
2	Goat shows smooth movement with its hind legs. It may raise a hind leg up to 5 cm from the floor or reposition its hind legs during milker contact.
3	The goat shows smooth, slow, and alternating leg movements. It may raise a hind leg from the floor up to 5 cm.
4	The goat shows vigorous, fast, and alternating movements with its hind or front legs, hitting the ground hard, which characterize “tapping”. It may not allow the teat cup to be fitted and needs to be restrained.
MOV Scores	
1	The goat does not exhibit frequent and vigorous head, ear, and tail movements. It may or may not be feeding.
2	The goat exhibits frequent head, ear, and tail movements, but not vigorous. It may or may not be feeding.
3	The goat exhibits vigorous and frequent head, ear, and tail movements. It is not feeding.

* Adapted from Paranhos da Costa and Broom [15].

## Data Availability

The data presented in this study are available on request from the corresponding author.

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
