# Peer review of "Milking Reactivity in Primiparous Saanen Goats During Early Lactation: Effects on Milk Yield, Milk Quality and Plasma Cortisol Concentration"

_animals, 2024, doi:10.3390/ani14233365_

Round 1
Reviewer 1 Report
Comments and Suggestions for Authors
The study aimed to investigate the progression of milking reactivity throughout the lactation period and the implications of milking reactivity of primiparous goats at the beginning of the lactation period on milk production and quality and plasma cortisol.
However, the results obtained by the authors were not consistent enough to draw a convincing conclusion. This finding can be confirmed by what is stated at the end of the discussion (lines 254-257): “Finally, our study has some limitations, and the main one is the limited number of animals available. Thus, we strongly recommend that further studies aiming to assess the effect of milking reactivity on milk yield and quality consider having a greater number of animals and expand the number of reactivity traits assessed.”
Another negative aspect of the research, which is highly relevant to explaining the data obtained, especially regarding milk production and composition, concerns the lack of information on dry matter consumption and nutrients that were not presented. Therefore, the differences detected in milk fat levels between the groups could not be explained.
Author Response
The study aimed to investigate the progression of milking reactivity throughout the lactation period and the implications of milking reactivity of primiparous goats at the beginning of the lactation period on milk production and quality and plasma cortisol.
However, the results obtained by the authors were not consistent enough to draw a convincing conclusion. This finding can be confirmed by what is stated at the end of the discussion (lines 254-257): “Finally, our study has some limitations, and the main one is the limited number of animals available. Thus, we strongly recommend that further studies aiming to assess the effect of milking reactivity on milk yield and quality consider having a greater number of animals and expand the number of reactivity traits assessed.”
Another negative aspect of the research, which is highly relevant to explaining the data obtained, especially regarding milk production and composition, concerns the lack of information on dry matter consumption and nutrients that were not presented. Therefore, the differences detected in milk fat levels between the groups could not be explained.
AU: Thank you for your comments. Despite the limitations of this study, the number of goats was sufficient to provide valuable results on the effects of reactivity on milking in primiparous goats, regardless of whether the results differed from those expected, which led to the rejection of the hypotheses that the more reactive goats in the milking parlor have a higher plasma cortisol concentration and somatic cell count and lower milk yield. Future studies with more animals may validate or challenge our findings, contributing, as in any scientific investigation, to improving the knowledge of the welfare of farm animals. We changed the conclusions to ensure more robustness (L261-267).
We agree that dry matter intake and nutrients influence milk production. Therefore, we ensured that all goats had access to the same diet, with the same amount of feed being provided to each goat to minimize potential variation caused by nutritional factors. This is now better reported in L105-107.

Reviewer 2 Report
Comments and Suggestions for Authors
In this manuscript, the authors have investigated the progression of the milking reactivity of primiparous goats over the lactation period and its implications at the beginning of the lactation period on milk yield and quality and plasma cortisol concentration. The results showed that SK and MOV scores reduced over the lactation period. Milk fat content was the only milk quality trait affected by milking reactivity, whereas there is no difference in milk yield, milk protein, and somatic cell count. The logic of the overall manuscript was clear. While I appreciate the effort of the work presented, I believe the authors needs to consider the following correction.
1.When animals are stimulated, they also secret an amount of adrenaline in addition to cortisol. While cortisol is certainly an important metric, what about adrenaline? Did the author measure the level of adrenaline in the blood? In addition, how does other hormones related to lactation performance such as prolactin change between less reactive and highly reactive goats?
2. Is there a reference basis for kicking behavior (SK) and head, ear, and tail movement (MOV) scoring? Please supplement it.
3. According to previous studies on cows, the consequence of calmer cows on milk fat, milk protein, and milk yield are inconsistent. Therefore, I think the effect of milking reactivity on milk quality seems meaningless, and it is not just a matter of animal numbers. How did the authors consider it?
4. SK and MOV score were identified on day 10, 30, and 60. Why did the authors only record milk yield, protein and fat contents, and somatic cell count of each goat on day 10 of the lactation period. Please explain it.
5. Please supplement the research significance in the abstract and introduction sections.
6. In figure 4, there are significant differences within the group, especially the less reactive group, so the reliability of the results needs to be considered.
7. Please indicate “n” value in the caption of each figure.
8. “P” should be italic in terms of specifications. Pleases revise it in the manuscript.
Comments on the Quality of English LanguageExtensive editing of English language required.
Author Response
In this manuscript, the authors have investigated the progression of the milking reactivity of primiparous goats over the lactation period and its implications at the beginning of the lactation period on milk yield and quality and plasma cortisol concentration. The results showed that SK and MOV scores reduced over the lactation period. Milk fat content was the only milk quality trait affected by milking reactivity, whereas there is no difference in milk yield, milk protein, and somatic cell count. The logic of the overall manuscript was clear. While I appreciate the effort of the work presented, I believe the authors needs to consider the following correction.
1.When animals are stimulated, they also secret an amount of adrenaline in addition to cortisol. While cortisol is certainly an important metric, what about adrenaline? Did the author measure the level of adrenaline in the blood? In addition, how does other hormones related to lactation performance such as prolactin change between less reactive and highly reactive goats?
AU: Thank you for your comment. It would be good to evaluate other physiological indicators. Catecholamine concentrations, including adrenaline, can be determined in plasma samples from several animal species, including goats, using radioenzymatic methods. However, to avoid physical stress from blood collection, the animals must have an intravenous catheter and be trained to collect blood, which was impossible to carry out under the conditions under which this study was conducted.
- Is there a reference basis for kicking behavior (SK) and head, ear, and tail movement (MOV) scoring? Please supplement it.
AU: There is no reference basis for SK and MOV in goats. We used the SK reactivity scores adapted from Paranhos da Costa and Broom (2001), who used similar variables to access cows reactivity during milking (L117 and L319-321), and we created the MOV score assessment.
- According to previous studies on cows, the consequence of calmer cows on milk fat, milk protein, and milk yield are inconsistent. Therefore, I think the effect of milking reactivity on milk quality seems meaningless, and it is not just a matter of animal numbers. How did the authors consider it?
AU: Yes, results with cows are indeed inconsistent, and this may also be the case with goats. Therefore, we encourage further studies to be conducted (L256-259).
- SK and MOV score were identified on day 10, 30, and 60. Why did the authors only record milk yield, protein and fat contents, and somatic cell count of each goat on day 10 of the lactation period. Please explain it.
A U: Figure 1 illustrates the variability in reactivity (SK and MOV) across the evaluated days. On day 10, the goats displayed higher reactivity values. By day 30, the scores had stabilized, and by day 60, most goats had habituated to the milking process, as indicated by lower SK and MOV values. Consequently, we focused on the implications of these differences in reactivity and milk production, particularly at the beginning of lactation (day 10), since it is a challenging phase for primiparous goats adapting to a new environment. Additionally, the number of less reactive and highly reactive animals was not balanced on days 30 and 60, making the comparisons between the groups difficult.
- Please supplement the research significance in the abstract and introduction sections.
AU: Thank you for your suggestion. The research significance was included in L34-35 and L79-82.
- In figure 4, there are significant differences within the group, especially the less reactive group, so the reliability of the results needs to be considered.
AU: In Figure 4, no significant differences were found between the groups. However, there was variation among individuals in both groups. This information was included L193-194 and L199.
- Please indicate “n” value in the caption of each figure.
AU: Thank you. Suggestion accepted (L168, L177, L188, L197).
- “P” should be italic in terms of specifications. Pleases revise it in the manuscript.
AU: Thank you. Corrected (L159, L171, L172, L178, L181-184, L190, L193).

Reviewer 3 Report
Comments and Suggestions for Authors
Review, paper no. animals-3242924 entitle „Progression of milking reactivity over the lactation period and its implications at the beginning of the lactation period on milk yield and quality and plasma cortisol concentration of primiparous Saanen goats”. The authors have used the standard journal format in manuscript writing. There are few minor observations that the authors should address before final submission. Please consider shortening the title.
Specific comments:
Abstract:
Is sufficiently presented (methods, general conclusions).
What relationships were found between cortisol concentration and reactivity ?
Introduction: The introduction section is sufficient and analytically and adequately covers the need for the study.
Line 82. Please add a research objective (please refer to the title of the work).
Methods: The methodology is sufficiently presented. However, it has a few inaccuracies.
What method was used to select animals for testing ?
Why are the goats milked once a day?
Is the presented (Table 1) assessment of MOV behavior the author's own?
What time after milking were the fat and protein content measured in milk? The same question concerns SCK.
Result and discussion
The results of the study are analytically presented. Figures are adequate explain the findings of the study.
Conclusion: In conclusion, generalizations are given. What were the relationships between reactivity and blood cortisol levels?
Author Response
Review, paper no. animals-3242924 entitle „Progression of milking reactivity over the lactation period and its implications at the beginning of the lactation period on milk yield and quality and plasma cortisol concentration of primiparous Saanen goats”. The authors have used the standard journal format in manuscript writing. There are few minor observations that the authors should address before final submission. Please consider shortening the title.
AU: Thank you. Suggestion accepted; we shortened the title (L2-4).
Specific comments:
Abstract:
Is sufficiently presented (methods, general conclusions).
What relationships were found between cortisol concentration and reactivity ?
AU: No effect of milking reactivity was observed on plasma cortisol concentration (L192); furthermore, the cortisol concentrations observed were low, indicating no stress for the goats in early lactation (L44-45).
Introduction: The introduction section is sufficient and analytically and adequately covers the need for the study.
Line 82. Please add a research objective (please refer to the title of the work).
AU: Thank you for your comment. The aim of this study was included in L84-86.
Methods: The methodology is sufficiently presented. However, it has a few inaccuracies.
What method was used to select animals for testing ?
AU: Thank you. No selection was carried out, the primiparous goats used in the study were those raised in the Animal Physiology Laboratory of the Faculty of Animal Science and Food Engineering, University of São Paulo, in Pirassununga, SP, Brazil (L94-96).
Why are the goats milked once a day?
AU: Yes (L99). According to the laboratory routine, the goats were milked once a day. Furthermore, it is common for goats to be milked only once a day because their daily milk production is not as high as that of dairy cows.
Is the presented (Table 1) assessment of MOV behavior the author's own?
AU: Yes. We (the authors) create the methodology for MOV assessment.
What time after milking were the fat and protein content measured in milk? The same question concerns SCK.
AU: The fat and protein content were measured immediately after milking. This information was included (L126), and the milk samples for SCC (not SCK) analysis were frozen and the analyses were performed later (L127-128).
Result and discussion
The results of the study are analytically presented. Figures are adequate explain the findings of the study.
AU: Thank you for your comment.
Conclusion: In conclusion, generalizations are given. What were the relationships between reactivity and blood cortisol levels?
AU: The milking reactivity did not affect the plasma cortisol concentration (L263=2-263).

Round 2
Reviewer 1 Report
Comments and Suggestions for Authors
Favorable for publication after corrections and considerations by the authors.